# Properties of Poplar Fiber/PLA Composites: Comparison on the Effect of Maleic Anhydride and KH550 Modification of Poplar Fiber

**DOI:** 10.3390/polym12030729

**Published:** 2020-03-24

**Authors:** Zhaozhe Yang, Xinhao Feng, Min Xu, Denis Rodrigue

**Affiliations:** 1Institute of Chemistry and Industry of Forest Products, Chinese Academy of Forestry, Nanjing 210042, Jiangsu, China; yzznefu@126.com; 2College of Furnishings and Industrial Design, Nanjing Forestry University, Nanjing 210037, Jiangsu, China; 3College of Material Science and Engineering, Northeast Forestry University, Harbin 150040, Heilongjiang, China; xumin1963@126.com; 4Department of Chemistry and Engineering, University of Laval, Quebec City, PQ G1V 0A6, Canada; denis.rodrigue@gch.ulaval.ca

**Keywords:** poplar fiber, polylactic acid, interfacial adhesion, KH550, maleic anhydride

## Abstract

To improve the interfacial adhesion and dispersion of a poplar fiber in a polylactic acid (PLA) matrix, maleic anhydride (MA) and a silane coupling agent (KH550) were used to modify the poplar fiber. The poplar fiber/PLA composites were produced with different modifier contents. The mechanical, thermal, rheological, and physical properties of composites were investigated. A comparison of different natural fiber modifications on the properties of composites was also analyzed. The results showed that both MA and KH550 could improve the interfacial adhesion between the poplar fiber and PLA, resulting in the enhanced mechanical properties of the composite, with 17% and 23% increases of tensile strength for 0.5% MA and 2% KH550, respectively. The thermal properties of the composites were improved at 6% KH550 (a 9% enhancement of *T_90%_*) and decreased at 0.5% MA (a 6% decrement of *T_90%_*). The wettability of the composites obtained a 11.3% improvement at 4% KH550 and a 5% reduction at 4% MA. Therefore, factors such as mechanical properties, economic efficiency, and durability should be carefully considered when choosing the modifier to improve the property of the composite.

## 1. Introduction

With the petroleum crisis growing, biocomposites (which are produced from natural resources), such as natural fiber-reinforced polyesters, have been extensively applied in structural and non-structural areas [1,2]. As a biocomposite, natural fiber-reinforced polylactic acid (PLA) has recently attracted great interest in composite science [3]. Natural fiber-reinforced PLA composites possess the advantages of both natural fibers and PLA, as natural fiber has a high strength, stiffness, and recyclability, and PLA is biodegradable due to its raw materials coming from the fermentation of renewable agricultural resources such as starch, sugar, and cellulose [4,5,6]. Furthermore, compared to the neat PLA, natural fiber-reinforced PLA composites with improved mechanical and thermal properties have been potentially used in engineering materials [1]. However, the different polarities between natural fibers and PLA is the major disadvantage of the composite, in which an inefficient interfacial layer is formed when the hydrophilic natural fiber is compounded with hydrophobic PLA [7]. This disadvantage results in the aggregation and uneven dispersion of natural fibers in the polymer matrix and the properties of composite decreased when compared to those of the neat polymer; moreover, it hinders the application of natural fiber-reinforced PLA composites in extended areas such as structure materials [8,9].

To improve the interfacial adhesion between natural fibers and PLA, modifications of natural fiber and PLA, separately or simultaneously, have been undertaken before raw material compounding [1,6,10,11,12]. PLA has been autoclaved to obtain different molecular weights of PLA, and its hydrophilicity was improved to result in a good mixability with hydrophilic materials, such as carbon nanotubes [13,14]. A compatibilizer has also been used to improve the compatibility of PLA with other polymers [15]. Compared to the complicated processes and high cost of PLA modification, natural fiber modification is more economic and feasible for composites production [6,16,17,18]. Generally, physical treatment (such as thermal treatment [19]), chemical modification (such as acylation), and coupling agents are used to modify natural fibers and improve their compatibility with polymers. Throughout all these methods, chemical modification, especially maleated and silane modification, has become a popular choice because of its beneficial results [1]. Maleic anhydride (MA) has been used to treat hemp fibers and the interfacial shear strength of composites with treated fiber increases by about 50% more than composites with untreated fibers [20]. The interfacial adhesion between bamboo fibers and PLA has been found to be significantly improved when bamboo fibers was treated by MA, resulting in the enhanced tensile properties and water resistance of the composite [21]. A silane coupling agent (such as KH550) has been commonly applied in the composite to improve the interfacial adhesion between fibers and the polymer through an ester-exchange reaction between the silane and hydroxyl groups of the fibers; additionally, the mechanical properties of composites have been found to be enhanced, and the serving life of composites has been found to be extended [1,22]. The performances of composite with maleated and a silane-modified natural fiber have been extensively studied, and the effect of modifier content on the properties of composites has also been investigated in previous studies [12,21,22,23,24,25]. However, few studies have been reported to compare the properties of natural fiber-reinforced composites between different chemical modifications of natural fibers.

Therefore, in this study, the poplar fiber was chemically-modified by different contents of MA and KH550, and the corresponding composites were produced. The comparison of different natural fiber modifications on the properties of composite was analyzed to investigate which modifier in what degree was optimal to treat the fiber and obtain superior properties of composite from the perspective of industrial production of composites. Additionally, the mechanical, thermal, and rheological properties and the wettability of the composites were analyzed.

## 2. Experimental

### 2.1. Materials 

PLA (4032D) with a density, melting temperature, glass transition temperature, and melting flow index of 1.25 g/cm^3^, 167 °C, 58 °C, and 1 g/min (210 °C/2.16 kg), respectively, was purchased from Nature Works (Minnetonka, MN, USA). Poplar fibers with average dimension of 293 × 62 μm^2^ (length × width) were produced from poplar wood (*Populus adenopoda*) in our lab. A silane coupling agent (γ-aminopropyltriethoxysilane, KH550), dicumyl peroxide (DCP), maleic anhydride (MA), acetone, ethanol, and acetic acid were supplied by Yongchang Chemical Company (Harbin, China). All chemicals were used as received.

### 2.2. Modification of Poplar Fiber

KH550 was hydrolyzed in the 95% ethanol solution for 40 min and mixed with acetic acid to adjust the solution pH to 3–4. Poplar fiber was added and magnetically stirred with KH550 in the solution for 10 min, then air-dried for 10 min and vacuum-dried at 80 °C for 20 min. The KH550-modified poplar fiber was obtained by activating the dried fiber at 120 °C for 2 h. The contents of KH550 (relative to the weight of the poplar fiber in the composite) were 2%, 4%, 6%, and 8%. The reaction scheme between KH550 and the poplar fiber is shown in Figure 1a.

The MA and DCP (used as the initiator) with a ratio of 10:1 were mixed and dissolved in 250 ml of acetone, and the mixed solution was evenly sprayed on the poplar fiber and dried at 80 °C for 2 h to volatilize the acetone. The grafting reaction between MA and the poplar fiber was taken at 110 °C for 2 h (Figure 1b). The MA contents relative to the weight of the poplar fiber in the composite were 0.5%, 1%, 2%, and 4%.

### 2.3. Preparation of Poplar Fiber/PLA Composites

Samples with 20% modified poplar fiber and 80% PLA were weighted and mixed in a high-speed mixer (SHR-10A, Zhangjiagang Tongsha Plastic Machinery Co., Ltd., Suzhou, Jiangsu, China) for 10 min and then compounded with a co-rotating twin-screw extruder (SJ45, Nanjing Rubber Machinery Factory, Nanjing, China) to produce pellets used in hot compression. The extruding temperature was 155 °C in the pumping zone, 165–175 °C in the melting zone, and 160 °C at the die; the extruding speed was 20 rpm. The pellets were compression-molded at 185 °C and 15 MPa for 4 min (SL-6, Harbin Special Plastics Products Co., Ltd., Harbin, China). Samples with 20% unmodified poplar fiber and 80% PLA were used as controls. The polymer and the poplar fiber (both modified and unmodified) were dried at 103 ± 2 °C for 12 h before composite preparation.

### 2.4. Characterizations

The chemical compositions of the KH550- and MA-modified poplar fibers were analyzed by Fourier transform infrared spectroscopy (Magna-IR 560, Thermo Nicolet, Ventura, CA, USA) at room temperature. Data were collected from 400 to 4000 cm^−1^ at a resolution of 4 cm^−1^ with 20 scans for each sample. Three replicates were taken for each condition.

The tensile, flexural, and impact strength of composite were measured with a universal mechanical tester (Regear, Shenzhen, China) with sample dimensions of 165 × 13 × 4 (L × T × R), 80 × 13 × 4, and 63.5 × 12.7 × 4 mm^3^, according to ASTM D638, ASTM D790, and ASTM D4812, respectively. The testing speed for tensile and flexural strength was 5 and 2 mm/min (with a span of 64 mm), respectively. Six replicates were taken for each condition.

The surface morphologies of the modified fiber and fractured composites were taken by an FEI Quanta 200 SEM (Hillsboro, OR, USA) at an accelerating voltage of 10 kV and a working distance of 8–12 mm. Before the image taking, samples were dried and gold-sputtered for 10 s.

The thermal stability of the composite was evaluated with a thermogravimetric analyzer (TGA, TA4, NETZSCH, Bavaria, Germany). The sample of about 5 mg was loaded in the aluminum crucible and heated at 10 °C/min from 40 to 600 °C under a nitrogen atmosphere (50 mL/min), and three replicates were measured for each condition.

The thermal performance of the composites was measured with a differential scanning calorimeter (DSC, Q20, TA Instruments, New Castle, DE, USA). Three temperature scans were taken during the test: Firstly, the sample of about 5 mg in the aluminum crucible was heated from 25 to 210 °C and held at 210 °C for 5 min to remove any residual moisture and erase the thermal history; then, it was cooled down to 25 °C and reheated to 210 °C. The heating and cooling rate were 10 °C/min, and the nitrogen flow was 20 mL/min. The temperature of the glass transition (*T_g_*), crystallization (*T_c_*), and melting (*T_m_*) of the composite were analyzed from the endothermic and exothermic peaks during second heating scan. The crystallization enthalpy (*ΔH_c_*) and the heat of fusion (*ΔH_m_*) were determined from the area of the melting peaks. The crystallinity (*X_c_)* was calculated according to Equation (1):(1)$$X_{C}\left(\%\right)=100\times \left(\Delta H_{m}-\Delta H_{C}\right)/\left(\Delta H_{f}\times W_{PLA}\right)$$
where *ΔH_f_* is the enthalpy of fusion for 100% crystalline PLA (93.6 J/g) [26] and *W_PLA_* is the weight fraction of PLA in the composite.

The storage modulus and loss factor (tan δ) of the composites were determined through dynamic mechanical analysis (DMA, Q800, TA Instruments, New Castle, DE, USA). Samples with dimensions of 55 × 13 × 4 mm^3^ (L × T × R) were heated at a rate of 3 °C/min from 35 to 150 °C. The frequency and strain during testing were 1 Hz and 0.1%, respectively, and three replicates were taken for each condition.

The dynamic rheological properties of the composites were evaluated with a rotational rheometer (Discovery HR-2, TA Instruments, New Castle, DE, USA). Before testing, the molten samples were equilibrated at 175 °C for 5 min to erase any previous thermal and deformational histories. A dynamic strain sweep was firstly taken to determine the linear viscoelastic region; then, a dynamic time sweep was conducted at an angular frequency of 6.283 rad/s and a strain of 0.01% for 3000 s to ensure that the frequency sweep remained in the linear range and without any degradation during testing. A dynamic frequency sweep (0.06–628.3 rad/s) was conducted at 190 °C and a strain amplitude of 0.01%. Three replicates were tested for each condition.

The surface wettability of the composite was measured with an OCA-20 (Dataphysics Instrument, Filderstadt, Germany) CA analysis system at room temperature. The water droplet volume was 5 μL, and the average contact angle of three measurements was collected.

## 3. Results and Discussions

### 3.1. Chemical and Structural Properties of Modified Poplar Fiber

The chemical components of the poplar fiber significantly changed after MA and KH550 modification (Figure 2a). For the MA-modified poplar fiber, the intensity at 1730 cm^−1^, which was caused by the carbonyl groups (C=O) in the poplar fiber, enhanced with the increasing of the MA content, and a new peak attributed to the carbon–carbon double bond (C=C) in MA was also observed at 1610 cm^−1^ [27]. These results confirmed that MA reacted with the hydroxyl group of the poplar fiber and was grafted onto the poplar fiber according to the modification scheme in Figure 1. A new peak at 1170 cm^−1^ appeared in the spectrum of the KH550-modified poplar fiber (Figure 2a), which was probably caused by the Si–O–C bonds formed between KH550 and the poplar fiber during high temperature activation [28]. Therefore, the chemical properties of the poplar fiber were chemically changed by the MA and KH550 modifications. Furthermore, these changes could be indirectly confirmed by the morphologies of the poplar fiber before and after modification (Figure 2b–e). Before modification, the surface of the poplar fiber was smooth, and little fiber tearing was observed (Figure 2b). However, after MA and KH550 modification, the surface of the poplar fiber became rough due to a layer of chemicals that was coated (Figure 2c,d), and fiber tearing was not found because the torn fiber might have been coated or broken into pieces during the reaction. With the increase of modifier content, the chemical layer on the poplar fiber became thicker, as shown in Figure 2d, and pit blocking was observed when 8% KH550 was used (Figure 2e).

### 3.2. Thermal Properties of Composites

The degradation performance of composites was measured by thermogravimetric analysis, and the temperatures of the initial (*T_5%_*, temperature at 5% weight loss), peak (*T_max_*, temperature at fastest weight loss), and end (*T_90%_*, temperature at 90% weight loss) degradation of MA- and KH550-modified composites were analyzed (Table 1). With the increase of MA content, all three degradation temperatures decreased, and these decrements demonstrated that the thermal stability of the MA-modified composites was declined (Figure 3a); of special note is that the decrease of *T_90%_* was more obvious than others. This might be explained by the promotion of the degradation of composites when a carboxyl group was introduced with the modified poplar fiber in the composites [29]. However, KH550 increased the degradation temperatures of the composites with increases in its content (Figure 3c), which mainly resulted from the reaction between KH550 and the poplar fibers, and a layer of KH550 on the fiber surface could have further improved the thermal stability of the composites [30].

After MA and KH550 modification, the glass transition temperature (*T_g_*) and crystallized temperature (*T_c_*) of the composites were barely affected, but the melting temperature (*T_m_*) irrelevantly increased with modifier content (Figure 3b, d; Table 2). From the crystallization enthalpy (*ΔH_c_*) and heat of fusion (*ΔH_m_*), the crystallinity of composites (*X_c_*) decreased when MA and KH550 were introduced. And the reduction of *X_c_* in KH550-modified composites was more obvious than MA, which was probably due to the fact that the reaction between KH550 molecules and the poplar fiber interrupted the forming of ordered hydrogen bonding between the poplar fiber and the PLA molecules, resulting in a decrease of *X_c_*. Free carboxyl groups still existed in the MA-modified poplar fiber, and the ordered hydrogen bonds between the poplar fiber and the PLA molecules could be replaced by those between carboxyl groups and the PLA molecules; therefore, the *X_c_* of the MA-modified composites remained.

### 3.3. Mechanical Properties of Composites

With increasing modifier content, the mechanical properties of both the MA- and KH550-modified poplar fiber/PLA composites had similar tendencies, as shown in Figure 4. The optimal mechanical properties were achieved at 0.5%–1% MA and 2%–4% KH550. The overall mechanical properties of the MA-modified composites were higher than those of the KH550-modified composites; however, the strengths of the 2% MA-modified composites were comparable with those of the 2%–4% KH550-modified composites. At a high modifier content, the impact strength and elongation of the 4% MA-modified composites significantly decreased due to unreacted carboxyl groups in the fiber that degraded the composite during processing [31]. The strengths were not obviously improved when 4%–8% KH550 was used, probably because the high steric hindrance of the KH550 molecules lowered the contact chance between the free KH550 molecules and hydroxyl groups on the poplar fiber [32], yielding high strengths of KH550-modified composites at high modifier content. The modulus of the composites increased with the increasing MA content and decreased with the increasing KH550 content. Of special note is that the flexural modulus increased by 9% at 1% MA, which might have been caused by the intact rigidity of the poplar fiber during MA modification, and the rigidity of composites could have been improved when rigid fiber was introduced [33]. However, a high KH550 content would probably have degraded the composite and decrease its rigidity and crystallinity; this can be certified by the crystallinity of composites that were analyzed in DSC measurements. Therefore, the modulus of the (4%–8%) KH550-modified composites was reduced. The elongation of the KH550-modified composites was improved (~57%) and leveled off at 4%–8% KH550, whereas the maximum increase of elongation was only 17% at 0.5% MA and a decrease was found when MA was further added (1%–4%).

The dynamic mechanical properties of the composites showed different behaviors between MA and KH550 modification (Figure 5). At a low temperature (<*T_g_*), the storage modulus of the KH550-modified composites was lower than that of unmodified composites; on the contrary, the MA-modified composites obtained a higher storage modulus than the unmodified composites. This was mainly because the hydrogen bond between the poplar fibers and the PLA was interfered with by KH550 molecules, which acted as plasticizers to improve the toughness of the composite (as increased elongation; see Figure 4f) [34,35] and promoted by MA modification in which the free carboxyl groups on the modified fiber could efficiently form hydrogen bonding with PLA [36]. Therefore, compared to unmodified composite, MA modification promoted the crystallinity of composites, and a decrease was found in the KH550-modified composites, resulting in obvious differences of the storage modulus between the MA- and KH550-modified composites at a low temperature. Above *T_g_*, a peak around 80–100 ℃ was found in both composites, which was mainly caused by the cold crystallization, in which the mobility of PLA molecule chains increased with increasing temperature, and the moving chains rearranged in order to form a crystal [37]. The crystallization was more obvious in the MA-modified composites than that of the KH550-modified composites, because the orientation ability of the molecule chain in the MA-modified composites was higher than that in the KH550-modified composites. Therefore, the storage modulus at the rubbery region increased for the MA-modified composites and decreased for the KH550-modified composites compared to that of the unmodified composites. The *T_g_* of the composites was characterized by a loss factor, as shown in Figure 5c,d, and it was hardly changed in the MA- and KH550-modified poplar fibers. This result was consistent with that measured by DSC analysis, as shown in Table 2.

### 3.4. Rheological Properties of Composites

Before the frequency scan, strain and time scans were performed (not shown) to guarantee that the strain of the composite was in the linear viscoelastic region and the degradation of the composite would not happen during frequency scan. Therefore, a strain of 0.01% and a time of 700 s were used in the frequency scan.

The storage modulus (G’) and loss modulus (G’’) of the unmodified composites fitted well with the relation of G’~ω and G’’~ω at a low frequency [38], respectively. The G’, G’’, and complex viscosity (η*) of the MA- and KH550-modified composites decreased compared to those of the unmodified composites (Figure 6). However, the value decrease of the MA-modified composites at a high frequency was lower than that of the KH550-modified composites, and the G’ and G’’ of the MA-modified composites were comparable with those of the unmodified composites when the scan frequency was above 600 rad/s. The maintained G’ and G’’ values of the MA-modified composites were probably caused by the improvement of resistance to shear deformation, which resulted from the oriented molecules formed by the free carboxyl groups with PLA molecules and the poplar fiber in the composite during high frequency shearing at high temperature [39]. The η* of the MA-modified composites remained stable at a low frequency (<10 rad/s) and significantly decreased with an increase of MA content at high frequency, indicating that MA acted as a plasticizer to diminish molecule entanglement in the composite, thus resulting in a low melting viscosity. The G’ and G’’ of the KH550-modified composites decreased with increasing KH550 content, and η* remained at a low frequency (<0.2 rad/s) and decreased at a high frequency (10–100 rad/s). This decrease was mainly caused by the plasticization effect of KH550, by which the melt fluidity of the composites was improved due to the hydrogen bonding between the PLA molecule and poplar fiber being reduced by KH550 [34].

### 3.5. Physical Properties of Composites

The physical properties, including the fractural morphology and wettability, of the composites were investigated. The interfacial adhesion between the poplar fiber and the PLA matrix was weak before fiber modification. Because gaps between the poplar fiber and the PLA matrix, and the smooth surface of the pulled poplar fiber were found, as shown in Figure 6a. After MA and KH550 modification, especially at 0.5% MA and 2% KH550, no obvious gap was found, and the poplar fiber was broken when pulled out from the PLA matrix (Figure 7b,c); these results demonstrate that the interfacial adhesion between the poplar fiber and the PLA matrix was significantly improved. At a high modifier content, the interfacial adhesion of 8% KH550 was similar to that of 2% KH550 (Figure 7d); however, a weak interfacial adhesion was found in the 4% MA-modified composites because of gaps in the matrix (Figure 7e). As mentioned before, the carboxyl groups degraded the poplar fiber during composite compounding and led to a weak adhesion between poplar fibers [31]; therefore, poor mechanical properties were obtained at 4% MA modification.

The wettability of the MA- and KH550-modified composites obtained opposite performances, as shown in Figure 8. With increase of KH550 content, the contact angle (CA) of the KH550-modified composites increased and leveled off between 4% and 8% KH550; however, the CA of the MA-modified composites decreased with increasing MA content. Compared to the CA of the unmodified composites (78°), maximally, about 11.3% of improvement was achieved at 4% KH550, and a 5% reduction was obtained at 4% MA; this was mainly due to the introduction of the hydrophobicity of KH550 and the hydrophilicity of MA in the composites, which changed the wettability of composites [38,39].

## 4. Conclusions

Both MA and KH550 modification could improve interfacial adhesion between poplar fibers and PLA, resulting in the enhancement of the mechanical properties of their composites. However, the enhancement of MA modification was more obvious than that of KH550 modification. The thermal properties of composites were increased by KH550 modification, and no change was found for MA modification. The crystallinity decreased in both composites, and the decrement of KH550 was higher than MA. For wettability, KH550 improved the hydrophobicity of the composites because of the increased CA, and more hydrophilicity of the composites was observed for MA modification due to the hydrophilicity of MA. Overall, KH550 was more effective than MA when all the properties were considered, especially at 2%–4%. However, MA could significantly and economically improve the mechanical properties of composites compared to those of KH550, because superior mechanical properties were obtained when only 0.5%–1% MA was used.

## Figures and Tables

**Figure 1 polymers-12-00729-f001:**
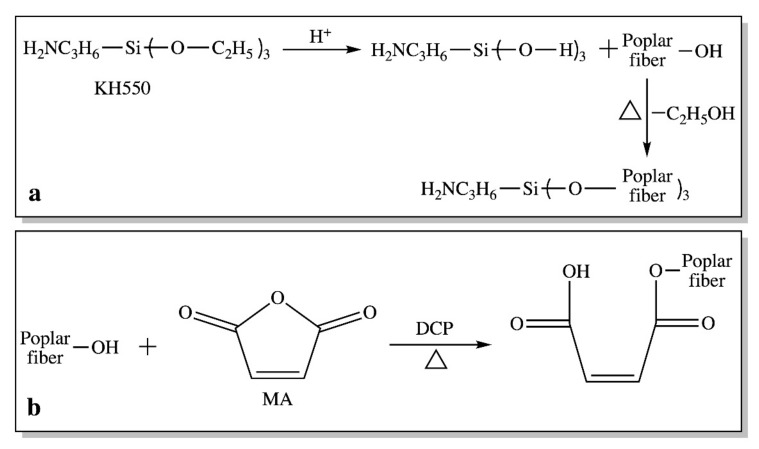
The poplar fiber modification scheme with a silane coupling agent (KH550) (**a**) and maleic anhydride (MA) (**b**).

**Figure 2 polymers-12-00729-f002:**
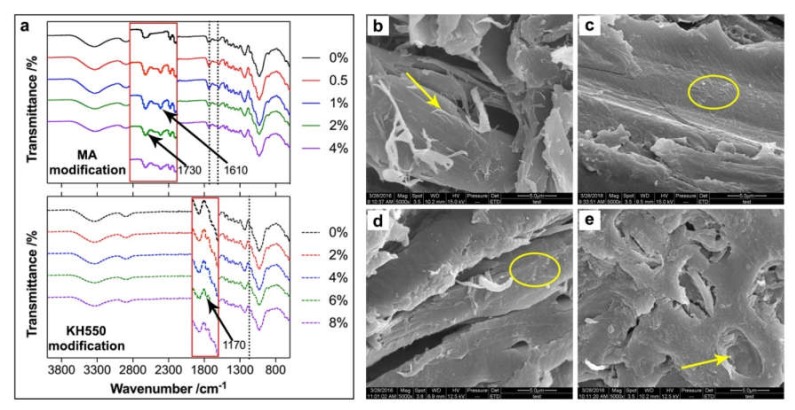
FTIR spectra (**a**) and morphologies (**b**–**e**) of the poplar fiber (b—unmodified; c—4% KH550-modified, e—8% KH550-modified; and d—4% MA-modified).

**Figure 3 polymers-12-00729-f003:**
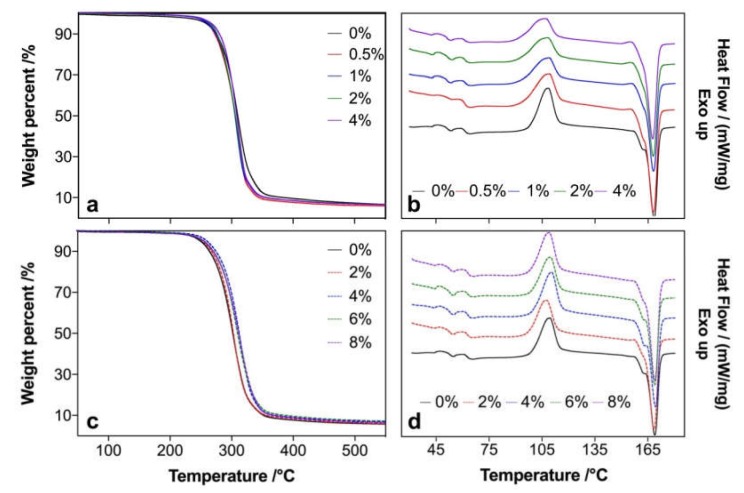
TG (**a**,**c**) and DSC (**b**,**d**) of MA- (**a**,**b**) and KH550- (**c**,**d**) modified poplar fiber/PLA composites.

**Figure 4 polymers-12-00729-f004:**
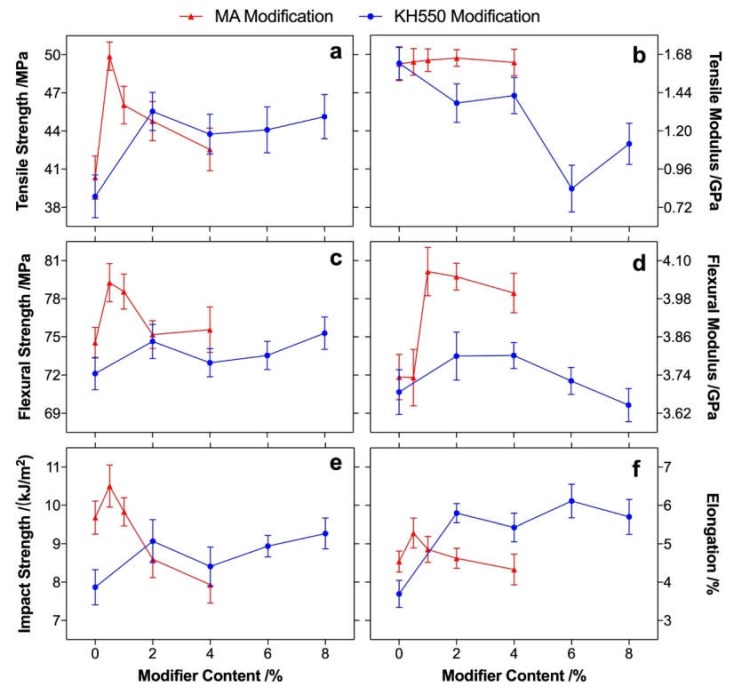
Tensile strength (**a**), tensile modulus (**b**), flexural strength (**c**), flexural modulus (**d**), impact strength (**e**), and elongation (**f**) of the MA- (triangle) and KH550- (circle) modified poplar fiber/PLA composites.

**Figure 5 polymers-12-00729-f005:**
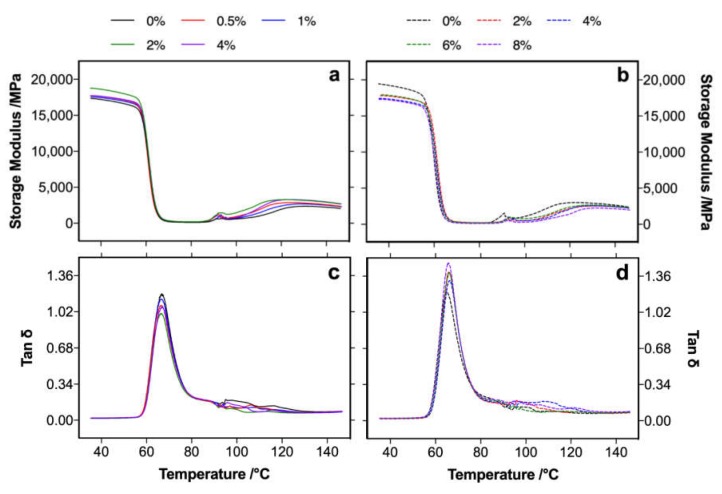
Storage modulus (**a**,**b**) and loss factor (**c**,**d**) of the MA- (**a**,**c**) and KH550- (**b**,**d**) modified poplar fiber/PLA composites.

**Figure 6 polymers-12-00729-f006:**
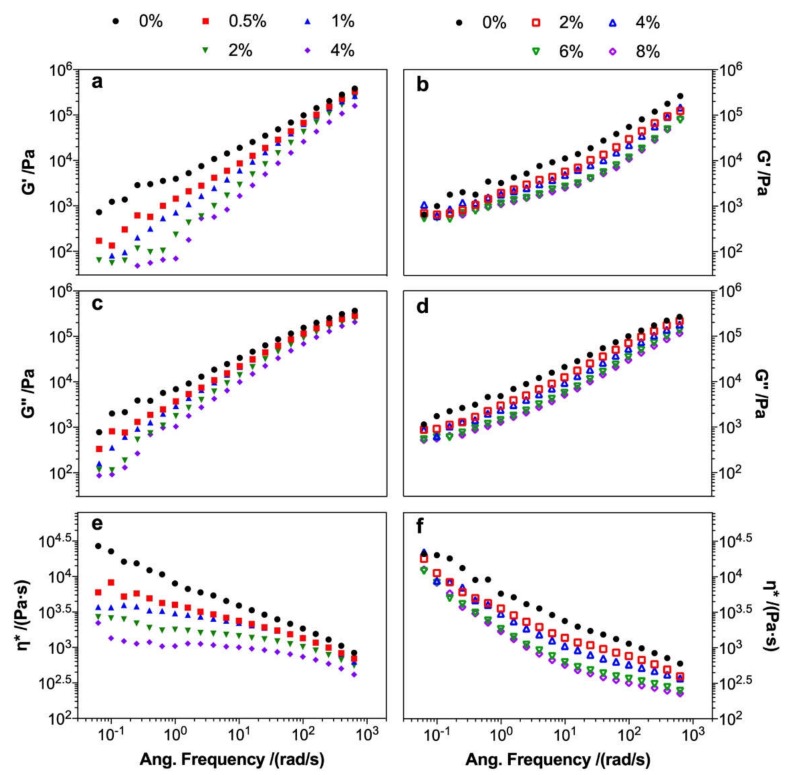
Rheological properties of the MA- (**a**,**c**,**e**) and KH550- (**b**,**d**,**f**) modified poplar fiber/PLA composites: (**a**,**b**) storage modulus, (**c**,**d**) loss modulus, and (**e**,**f**) complex viscosity.

**Figure 7 polymers-12-00729-f007:**
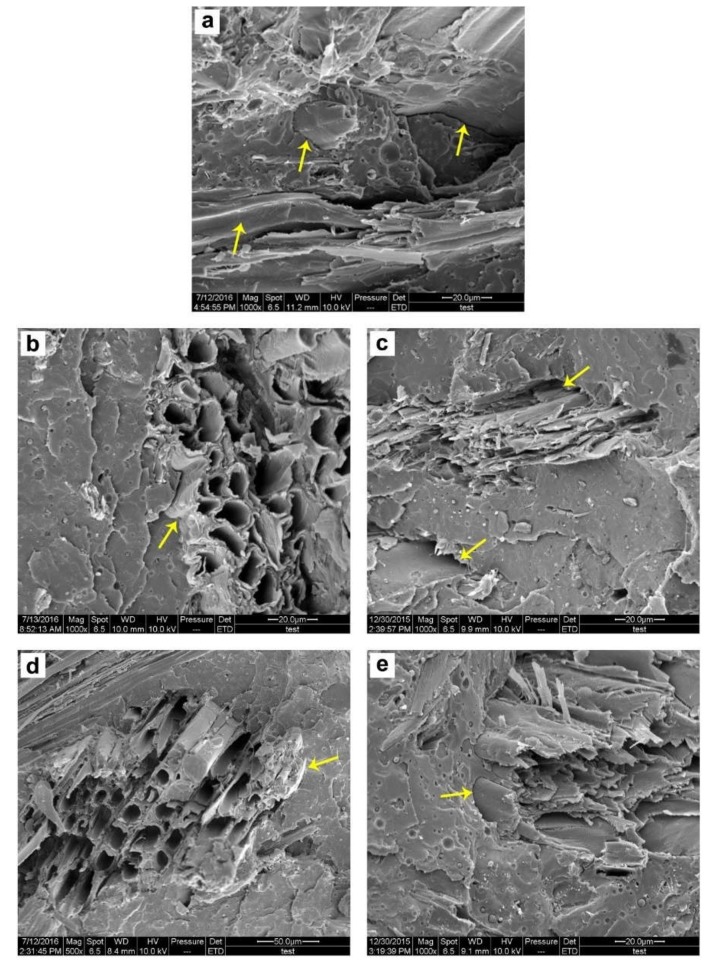
The morphology of the PLA composites with unmodified poplar fiber (**a**), 2% KH550-modified poplar fiber (**b**), 8% KH550-modified poplar fiber (**d**), 0.5% MA-modified poplar fiber (**c**), and 4% MA-modified poplar fiber (**e**).

**Figure 8 polymers-12-00729-f008:**
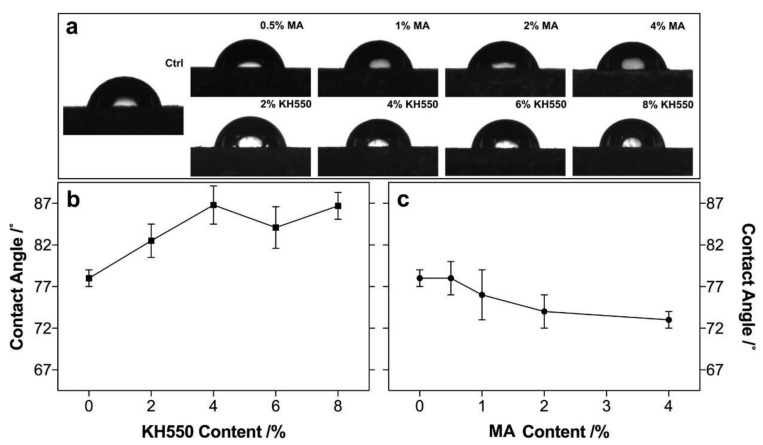
The contact angle morphology (**a**) and value (**b**,**c**) of the PLA composites with the KH550-modified poplar fiber (**b**) and the MA-modified poplar fiber (**c**).

**Table 1 polymers-12-00729-t001:** Temperatures of initial (*T_5%_*), peak (*T_max_*) and end (*T_90%_*) degradation of MA- and KH550-modified poplar fiber/PLA (polylactic acid) composites.

Modifier Content /%		*T_5%_/*°C	*T_max_/*°C	*T_90%_/*°C
	0	250	301	352
MA	0.5	251	299	333
	1	251	300	340
	2	253	300	340
	4	255	302	343
KH550	2	253	305	355
	4	259	317	366
	6	254	309	381
	8	256	316	369

**Table 2 polymers-12-00729-t002:** Differential scanning calorimeter (DSC) thermal performance of MA- and KH550-modified poplar powder/PLA composites.

Modifier Content /%		*T_g_/*°C	*T_c_/*°C	*T_m_/*°C	*ΔH_c_/(J/g)*	*ΔH_m_/(J/g)*	*X_c_/%*
	0	61.7	108.5	167.5	20.4	31.1	37.7
MA	0.5	61.9	109.1	168.2	22.0	35.3	35.2
	1	61.4	108.7	167.8	19.1	29.7	36.2
	2	61.6	107.9	167.9	19.5	29.5	36.3
	4	61.3	106.9	167.7	19.6	29.2	34.9
KH550	2	61.5	106.9	168.3	20.4	30.6	34.9
	4	61.4	109.4	168.9	22.6	31.0	31.4
	6	61.8	108.9	168.5	21.2	30.3	32.1
	8	61.7	108.3	168.8	23.8	32.8	33.7

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
