# Peer review of "Properties of Poplar Fiber/PLA Composites: Comparison on the Effect of Maleic Anhydride and KH550 Modification of Poplar Fiber"

_polymers, 2020, doi:10.3390/polym12030729_

Round 1
Reviewer 1 Report
The subject of the article are the PLA-based composites filled with wood flour modified in different ways. The paper is interesting but in my opinion some issues need to be corrected before publication in the Polymers journal. Below please find my comments:
- Line 47: Is heating of the fibers actually used in order to improve its compatibility with the polymer? Please add a reference.
- 1. Materials. Is there any specific reason why poplar wood flour was chosen? If poplar wood flour has any interesting properties that can be advantageous in polymeric composites, it would be good to include it in the paper.
- Line 76: please include the full chemical name of the KH550 silane
- Line 82:how long was mixing of the wood flour with the silane solution?
- Line 89: Do I understand correctly, that the grafting reaction at 110C was conducted after drying the wood flour?
- 3. Preparation of poplar fiber/PLA composites: was the polymer and the filler dried before processing? If yes, add the drying conditions. What was the poplar fiber content in the composites? What’s more, the extrusion temperature is rather low, taken into consideration that the melting temperature of the composite was about 168C.
- Lines 115 and 119: please add crucible types that were used in the DSC and TGA experiments.
- Formula (1): the expression ΔHfxWpla should be presented in parenthesis: (ΔHfxWpla)
- Line 131: The 38C/min heating rate is very high. Isn’t it a typo? What’s more, add the strain (amplitude) value during the DMA experiment
- Results and Discussion: The discussion of the properties would be easier to follow if pure PLA was used as a reference. If this is impossible, please add the fiber content, because it greatly influences the properties of the composites.
- Line 153: The FTIR spectrum presented in Figure 2 does not directly imply a change in the fibers’ hydrophilicity. The –OH peak around 3300 cm-1 does not change.
- Figure 2: The differences in the intensity of 1730, 1610 and 1170 peaks are hard to spot in this small graph. I recommend you to change the scale in this graph.
- Line 182: You refer to the changes in the composites’ crystallinity before discussing the results of DSC measurements. I would recommend to rearrange the article and put the DSC results before the discussion of the mechanical properties.
- Line 194: The chemical interactions between the polymer and the modified fibers should be examined e. g. by FTIR in order to check the claim that the KH550 interferes the hydrogen bonds between the fiber and PLA.
- Figure 4: The Y axis could be in logarithmic scale in order to make the differences between the different samples more visible.
- Line 222: I think that it would be interesting to also analyze the thermal degradation of the modified fibers themselves.
- Line 243 the relation G’-ω2 is not presented. I suppose you meant the G’-ω relationship.
- Figure 5: The results for the composite with the unmodified fiber should be marked with the same symbols to avoid any confusion.
Author Response
Responses for reviewer 1’s comments
(polymers-748174)
Dear Reviewer 1,
We thank for your conscientious comments that help us improve the quality of our manuscript. All the comments have been replied point by point as shown below and a careful revision to the manuscript is done accordingly. All changes were made in RED color in the manuscript.
- Line 47: Is heating of the fibers actually used in order to improve its compatibility with the polymer? Please add a reference.
Response: Yes, the number of hydroxyl groups and the polarity of the fibers decreased after heating at high temperature, resulting in good compatibility with the polymer. We have changed “heating” into “thermal treatment” and added some references in the manuscript.
- Is there any specific reason why poplar wood flour was chosen? If poplar wood flour has any interesting properties that can be advantageous in polymeric composites, it would be good to include it in the paper.
Response: Because the poplar wood is one of fast-growing wood species in China and its properties are not good enough when directly applied it into wood products. Pulverizing it into wood flour is one of efficient way to use it.
- Line 76: please include the full chemical name of the KH550 silane
Response: We have added the full name of the KH550 silane in the manuscript as follows: “Silane coupling agent (γ-aminopropyltriethoxysilane, KH550)…”.
- Line 82: how long was mixing of the wood flour with the silane solution?
Response: It is 10 min. We added the mixing time in the manuscript as follows: “Poplar fiber was added and magnetic stirred with KH550 in the solution for 10 min…”
- Line 89: Do I understand correctly, that the grafting reaction at 110 °C was conducted after drying the wood flour?
Response: Yes. The grafting reaction was conducted at 110 °C after drying the woof flour at 80 °C.
- Preparation of poplar fiber/PLA composites: was the polymer and the filler dried before processing? If yes, add the drying conditions. What was the poplar fiber content in the composites? What’s more, the extrusion temperature is rather low, taken into consideration that the melting temperature of the composite was about 168 °C.
Response: We have added the dry condition of the polymer and poplar fiber as follows: “Samples with 20% modified poplar fiber and 80% PLA were… The polymer and poplar fiber (modified and unmodified) were dried at 103±2 ℃ for 12 h before composites preparation.”.
We have rechecked the temperature, and corrected in the manuscript as follows: “The extruding temperature was 155 ℃ in the pumping zone, 165-175 ℃ in the melting zone and 160 ℃ at the die…”.
- Lines 115 and 119: please add crucible types that were used in the DSC and TGA experiments.
Response: Thanks reviewer’s suggestion. We have added the crucible types (aluminum crucible) in the DSC and TGA experiments.
- Formula (1): the expression ΔHfxWpla should be presented in parenthesis: (ΔHfxWpla)
Response: Thanks reviewer’s suggestion. We have moved the expression into parenthesis.
- Line 131: The 38 °C /min heating rate is very high. Isn’t it a typo? What’s more, add the strain (amplitude) value during the DMA experiment
Response: It was our mistake. The heating rate has been changed into “3 °C/min”, and the strain used in the measurement was also added as follows: “The frequency and strain during testing was 1 Hz and 0.1%, respectively…”.
- Results and Discussion: The discussion of the properties would be easier to follow if pure PLA was used as a reference. If this is impossible, please add the fiber content, because it greatly influences the properties of the composites.
Response: We have added the fiber content in the section 2.3 as follows: “Samples with 20% modified poplar fiber and 80% PLA were weighted…”
- Line 153: The FTIR spectrum presented in Figure 2 does not directly imply a change in the fibers’ hydrophilicity. The –OH peak around 3300 cm-1 does not change.
Response: We have changed the statement into “…the chemical properties of poplar fiber have been chemically changed by MA and KH550 modification…”.
- Figure 2: The differences in the intensity of 1730, 1610 and 1170 peaks are hard to spot in this small graph. I recommend you to change the scale in this graph.
Response: We have replaced the Figure 2 with insert enlarged FTIR spectrums.
- Line 182: You refer to the changes in the composites’ crystallinity before discussing the results of DSC measurements. I would recommend to rearrange the article and put the DSC results before the discussion of the mechanical properties.
Response: Thanks reviewer’s suggestion. We have rearranged the article and put the DSC results before the discussion of the mechanical properties.
- Line 194: The chemical interactions between the polymer and the modified fibers should be examined e. g. by FTIR in order to check the claim that the KH550 interferes the hydrogen bonds between the fiber and PLA.
Response: Actually, we did the FTIR examination for the composites, however, evidence was not found for our speculation. But we will systematically study the chemical interaction between polymer and modified fiber by other characterizations e.g. NMR in our next publication.
- Figure 4: The Y axis could be in logarithmic scale in order to make the differences between the different samples more visible.
Response: We redrew the Figure 4 (below) in logarithmic scale, it did make the difference at high temperature more visible, however, the changed the difference at low temperature. Therefore, we did not replace the Figure 4 with this new one.
Figure 4
- Line 222: I think that it would be interesting to also analyze the thermal degradation of the modified fibers themselves.
Response: Generally, the thermal stability of wood fiber is lower than its composites (Liu, R. et al. Compos Sci Tech, 2018, 103, 1-7). Therefore, it is reasonable for that decrease of thermal stability of modified wood fiber/PLA composites in our study, furthermore, the existence of carboxyl groups in the modified fiber can accelerate the decomposing of composites, which can be certified by previous work (Cervantes-Uc, J. et al. Polym Degrad Stabil 2006, 91, 3312-3321).
- Line 243 the relation G’-ω2is not presented. I suppose you meant the G’-ω relationship.
Response: We have checked the relationship between G’ and ω, it should be G’~ω, we have corrected in the manuscript.
- Figure 5: The results for the composite with the unmodified fiber should be marked with the same symbols to avoid any confusion.
Response: Thanks reviewer’s suggestion. We have changed the mark into same symbols.
Reviewer 2 Report
Yang et al. reported the manuscript entitled “Properties of Poplar Fiber/PLA Composites: Comparison on the Effect of Maleic Anhydride and KH550 Modification of Poplar Fiber’’ with detailed mechanical, thermal, rheological properties, morphology and contact angle. This manuscript needs more revision before publication.
- Abstract should be more quantitative.
- In the introduction part, the novelty of this work should be highlighted in the last paragraph.
- The introduction section should be more informative with some PLA-based articles. The author should introduce the following papers.
- Thermal properties, phase morphology and stability of biodegradable PLA/PBSL/HAp composites.
- Characterization of poly(lactic acid)s with reduced molecular weight fabricated through an autoclave process.
- Enhanced thermal stability, toughness, and electrical conductivity of carbon nanotube-reinforced biodegradable poly(lactic acid)/poly(ethylene oxide) blend-based nanocomposites
- In line 54, authors need to check carefully. I think water resistance not waster resistance.
- In the DMA part, the author stated ‘‘storage modulus of KH550 modified composites was lower than that of unmodified composites’’. Line 194, The author mentioned ‘‘this is mainly because the hydrogen bond between poplar fiber and PLA was interfered by KH550 molecules’’. Do you have any evidence for this statement. The authors should explain clearly why decreased?
- Authors need to provide XRD patterns of the samples.
- The author should give DSC cooling and melting curves.
- Regarding TGA part, the author should give TGA degradation curves of the prepared samples. In Table 1, Why T5% of modified 1 wt.% loading showed lower value compared to the 0.5 wt.% MA loading? Please mention the decomposition temperature in round figure (250 °C not 250.2 °C).
Author Response
Responses for reviewer 2’s comments
(polymers-748174)
Dear Reviewer,
We thank for your conscientious comments that help us improve the quality of our manuscript. All the comments have been replied point by point as shown below and a careful revision to the manuscript is done accordingly, and the English of our manuscript was also proofread. All changes were made in RED color in the manuscript.
- Abstract should be more quantitative.
Response: We have quantified the abstract as follows: “…resulting in enhanced mechanical properties of the composite, especially 17% and 23% increase of tensile strength for 0.5% MA and 2% KH550, respectively. The thermal properties of composites were improved at 6% KH550 (9% enhancement of T90%) and decreased at 0.5% MA (6% decrement of T90%). The wettability of composites obtained 11.3% improvement at 4% KH550 and 5% reduction 4% MA…”
- In the introduction part, the novelty of this work should be highlighted in the last paragraph.
Response: We have rearranged the novelty of our work in the last paragraph as follows: “Therefore, in this study, poplar fiber was chemical modified by different contents of MA and KH550, respectively, and the corresponding composites were produced. The comparison of different natural fiber modifications on the properties of composite was analyzed to investigate that which modifier and in what degree of it is optimal to treat the fiber and obtain the superior properties of composite from the perspective of industrial production of composites. Also, the mechanical, thermal, and rheological properties and wettability of composite were analyzed.”
- The introduction section should be more informative with some PLA-based articles. The author should introduce the following papers.
Response: We have added more information about PLA modifications in the manuscript as follows: “PLA was autoclaved to obtain different PLA molecular weight and its hydrophilicity was improved resulting in good mixability with hydrophilic materials, such as carbon nanotube [13,14]. Compatibilizer was also used to improve the compatibility of PLA with other polymers [15].” And the papers reviewer suggested were referred in the manuscript as “[13], [14], and [15]”.
- In line 54, authors need to check carefully. I think water resistance not waster resistance.
Response: Thanks reviewer’s reminding. We have had a careful recheck through the manuscript to avoid the mistake like this.
- In the DMA part, the author stated ‘‘storage modulus of KH550 modified composites was lower than that of unmodified composites’’. Line 194, The author mentioned ‘‘this is mainly because the hydrogen bond between poplar fiber and PLA was interfered by KH550 molecules’’. Do you have any evidence for this statement? The authors should explain clearly why decreased?
Response: We have added some references and explained the statement in the manuscript as follows: “This is mainly because the hydrogen bond between poplar fiber and PLA was interfered by KH550 molecules which acts as a plasticizer to improve the toughness of composite (as increased elongation shown in Figure 4f) [34, 35]…”
- Authors need to provide XRD patterns of the samples.
Response: We did the XRD patterns of the composites, however, no difference was found between those samples, especially between MA modified samples. This probably because crystallinity from XRD is not as precise as DSC in which the crystallization enthalpy was precisely collected to calculate the crystallinity (Liu, X. et al. Eur Polym J, 2002, 38, 1383-1389). Especially, this phenomenon is more obvious when most of components of the testing sample are same (20% wood fiber and 80% PLA). Therefore, we did not perform the XRD in our manuscript.
- The author should give DSC cooling and melting curves.
Response: We have inserted the DSC curves as “Figure 3b and d” in the manuscript.
- Regarding TGA part, the author should give TGA degradation curves of the prepared samples. In Table 1, Why T5%of modified 1 wt.% loading showed lower value compared to the 0.5 wt.% MA loading? Please mention the decomposition temperature in round figure (250 °C not 250.2 °C).
Response: We have given the TGA curves as “Figure 3a and c” in the manuscript. We have retested the sample and gotten the similar values of T5% between 0.5% and 1%, and the new value was updated. This variation is probably due to the uneven dispersion of the components in composites during hot compression. The decomposition temperature in table 1 has been changed into round figure.
Round 2
Reviewer 1 Report
I would like to thank the Authors for including the several comments in order to improve the manuscript, as well as suitable comment. In my opinion the manuscript may be published in presented form.
Reviewer 2 Report
Accept.